# Population genomics provides insights into the evolution and adaptation to humans of the waterborne pathogen *Mycobacterium kansasii*

Tao Luo [1,2,17✉], Peng Xu[2,3,17], Yangyi Zhang[4], Jessica L. Porter[5,6], Marwan Ghanem[7], Qingyun Liu [2], Yuan Jiang [4], Jing Li[4], Qing Miao[8], Bijie Hu[8], Benjamin P. Howden [5,6,9], Janet A. M. Fyfe[10], Maria Globan[10], Wencong He[11], Ping He[11], Yiting Wang[11], Houming Liu[12], Howard E. Takiff[13,14,15], Yanlin Zhao [11✉], Xinchun Chen [16✉], Qichao Pan [4✉], Marcel A. Behr [7✉], Timothy P. Stinear [5,6✉] & Qian Gao [2✉]

*Mycobacterium kansasii* can cause serious pulmonary disease. It belongs to a group of closely-related species of non-tuberculous mycobacteria known as the *M. kansasii* complex (MKC). Here, we report a population genomics analysis of 358 MKC isolates from worldwide water and clinical sources. We find that recombination, likely mediated by distributive conjugative transfer, has contributed to speciation and on-going diversification of the MKC. Our analyses support municipal water as a main source of MKC infections. Furthermore, nearly 80% of the MKC infections are due to closely-related *M. kansasii* strains, forming a main cluster that apparently originated in the 1900s and subsequently expanded globally. Bioinformatic analyses indicate that several genes involved in metabolism (e.g., maintenance of the methylcitrate cycle), ESX-I secretion, metal ion homeostasis and cell surface remodelling may have contributed to *M. kansasii*'s success and its ongoing adaptation to the human host.

[1] Department of Pathogen Biology, West China School of Basic Medical Sciences & Forensic Medicine, Sichuan University, Chengdu, China. [2] Shanghai Institute of Infectious Disease and Biosecurity, Key Laboratory of Medical Molecular Virology (MOE/NHC/CAMS), Shanghai Medical College and School of Basic Medical Sciences, Shanghai Public Health Clinical Center, Fudan University, Shanghai, China. [3] Key Laboratory of Characteristic Infectious Disease & Bio-safety Development of Guizhou Province Education Department, Institute of Life Sciences, Zunyi Medical University, Zunyi, China. [4] Department of Tuberculosis Control, Shanghai Municipal Centre for Disease Control and Prevention, Shanghai, China. [5] Department of Microbiology and Immunology, Doherty Institute for Infection and Immunity, University of Melbourne, Melbourne, Vic, Australia. [6] Doherty Applied Microbial Genomics, Doherty Institute for Infection and Immunity, University of Melbourne, Melbourne, Vic, Australia. [7] Department of Microbiology and Immunology, McGill University and McGill International TB Centre, Montreal, Quebec, Canada. [8] Department of Infectious Diseases, Zhongshan Hospital, Fudan University, Shanghai, China. [9] Microbiological Diagnostic Unit Public Health Laboratory, Doherty Institute for Infection and Immunity, University of Melbourne, Melbourne, Victoria 3000, Australia. [10] Victorian Infectious Diseases Reference Laboratory, Doherty Institute for Infection and Immunity, Melbourne Health, Melbourne, Vic, Australia. [11] Chinese Center for Disease Control and Prevention and Beijing Tuberculosis and Thoracic Tumor Research Institute, Beijing, China. [12] Department of Clinical Laboratory, The Third People's Hospital of Shenzhen, Southern University of Science and Technology, Shenzhen, China. [13] Unité de Pathogenetique Integrée Mycobacterienne, Institut Pasteur, Paris, France. [14] Laboratorio de Genética Molecular, CMBC, IVIC, Caracas, Venezuela. [15] Shenzhen Nanshan Center for Chronic Disease Control, Shenzhen, China. [16] Guangdong Provincial Key Laboratory of Regional Immunity and Diseases, Department of Pathogen Biology, Shenzhen University School of Medicine, Shenzhen, China. [17] These authors contributed equally: Tao Luo, Peng Xu. ✉email: taoluo@scu.edu.cn; zhaoyl@chinacdc.cn; chenxinchun@szu.edu.cn; panqichao@scdc.sh.cn; marcel.behr@mcgill.ca; tstinear@unimelb.edu.au; qiangao@fudan.edu.cn

Nontuberculous mycobacteria (NTM) are environmental bacteria, but some species can cause opportunistic infections in humans. While they are not as pathogenic as *Mycobacterium tuberculosis*, diseases due to NTM have been an increasing concern in global health[1–4], and in some developed countries, NTM is now responsible for more diseases than *M. tuberculosis*[2,4]. *Mycobacterium kansasii* is among the most pathogenic NTM and has the highest clinical relevance[5]. It is one of the last species to have diverged from a common ancestor before the appearance of the *M. tuberculosis* complex[6] and is capable of causing aggressive and destructive pulmonary disease resembling tuberculosis[7]. In the mid-20th century, before the emergence of the HIV pandemic, *M. kansasii* was dominant among NTM diseases in several regions of United States, Europe, and Japan[3]. It is currently one of the most frequent causes of NTM pulmonary disease throughout the world (Fig. 1a,

Supplementary Table 1), with a relatively high incidence in regions of Europe, South America, Africa, and Asia[3,8]. In China, *M. kansasii* has been isolated from pulmonary infections in many areas, but the incidence is highest in the highly urbanized eastern and southern coastal regions[9–12]. From 2008 to 2012 in Shanghai, it was responsible for nearly half of all NTM infections[13].

As with other NTM, *M. kansasii* infections are generally assumed to be acquired from environmental sources rather than by human-to-human transmission. Although municipal water distribution systems are believed to be the major reservoir for human *M. kansasii* infections[3,5,14], water isolates are usually genetically distinct from clinical strains. Molecular typing has revealed that *M. kansasii* comprises at least six distinct subtypes that vary in prevalence and clinical relevance[15–18]. Very recently, based on genome-wide average nucleotide identity (gANI), it was proposed that the subtypes should more accurately be designated

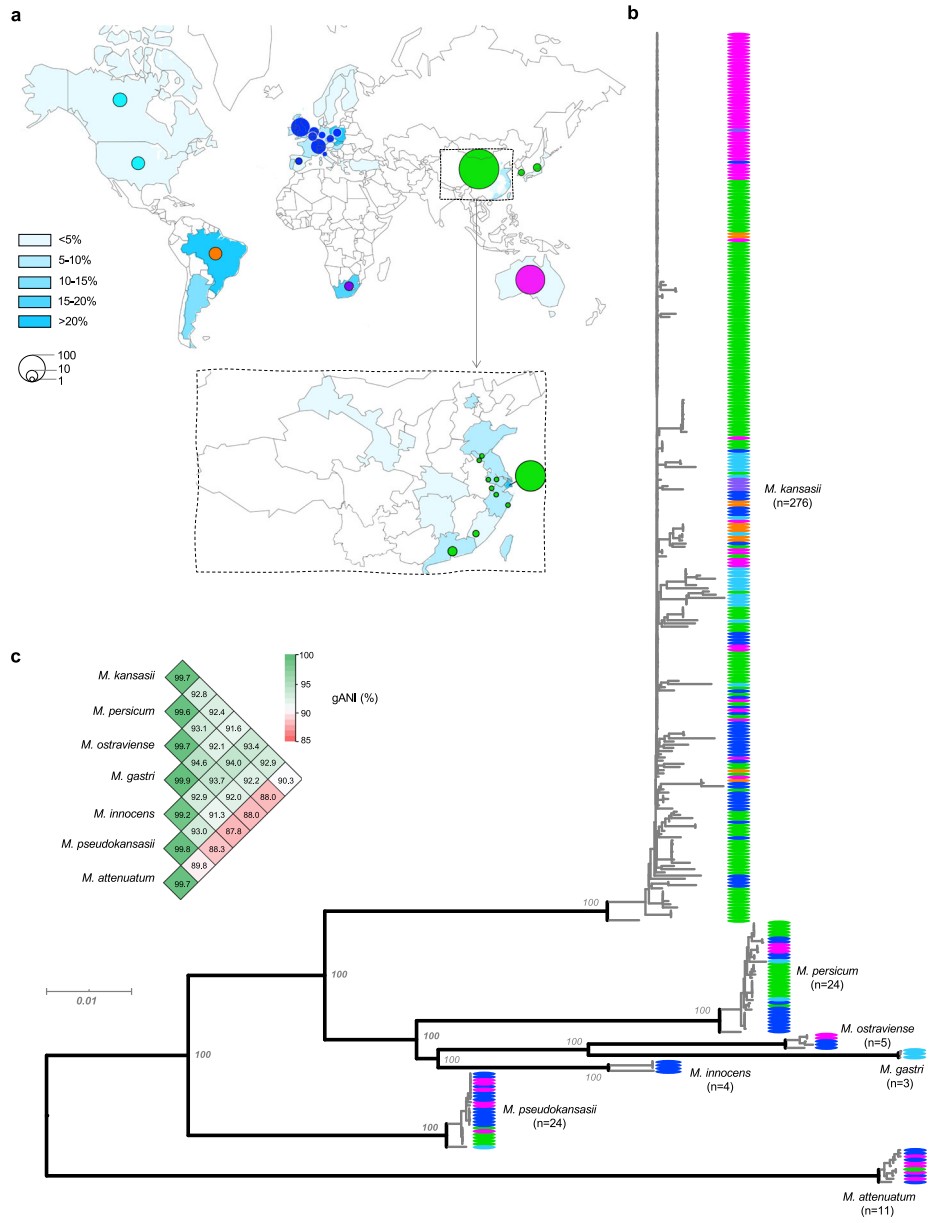

**Fig. 1 Global diversity of the *M. kansasii* complex. a** Geographical distribution of the 358 isolates in the study. The gradient blue colors indicate the prevalence of *M. kansasii* among NTM disease. **b** Core genome-based maximum-likelihood phylogeny of the 358 isolates. The colors of the terminal nodes correspond to the geographical origin of individual isolates, as denoted by the circles in (**a**). **c** Pairwise genomic average nucleotide identity (gANI) within and between the *M. kansasii* complex species. Source data are provided as a Source Data file.

as closely related species[19,20]. The six subtypes were designated as *M. kansasii* (former subtype I), *Mycobacterium persicum* (II), *Mycobacterium pseudokansasii* (III), *Mycobacterium ostraviense* (IV), *Mycobacterium innocens* (V), and *Mycobacterium attenuatum* (VI), which together with *Mycobacterium gastri*, were recognized as the *M. kansasii* complex (MKC)[19,20]. *M. kansasii* (former subtype I) is responsible for the vast majority of infections due to the MKC species worldwide but is not often isolated from water sources[15,17,18], and no definitive epidemiological link has ever been established between water reservoirs and clinical *M. kansasii* infections[21,22]. Instead, genotyping has shown that clinical strains of *M. kansasii* isolated from diverse geographic locations constitute a homogenous population[15,18], suggesting potential human-to-human transmission of a successful clone. Potential transmission of *M. kansasii* between family members has been reported in several cases[23,24]. In addition, transmission has been recently revealed as a major route for the dissemination of dominant clones of *Mycobacterium abscessus*[25], another NTM that can cause pulmonary infection.

Consistent with its clinical dominance, *M. kansasii* has the highest clinical relevance among the MKC, as it has been associated with severe and even fatal disease in both immune-competent and immune-compromised patients, while the other MKC species are isolated only from immune-compromised patients or environmental sources[17]. Although *M. kansasii* causes more disease than the other MKC species, the genetic determinants of its pathogenic adaptation have not been addressed. In addition, clinical *M. kansasii* isolates can vary phenotypically, with strains showing either a smooth or rough colony morphology due to differences in cell wall hydrophobicity[26,27]. Similar to *M. abscessus*[28], *M. kansasii* strains with the rough colony appear to be more virulent and can establish chronic systemic infections in mice[27,29], but the genetic basis for the phenotypic differences has not been explained.

In the current study, we analyzed the genomes of a worldwide collection of isolates to better define the global population structure of *M. kansasii*. The genomic analyses provided insights into its speciation, diversification, the sources of clinical infections, and possible genetic determinants associated with its ability to proliferation and cause disease in humans.

## Results

**Global diversity of the MKC.** We performed whole-genome sequencing on 271 MKC isolates, including 155 isolates from China, 74 isolates from Australia, 35 clinical isolates from European and North American countries, 5 from South Africa, and 3 from Japan. These genomes, together with an additional 86 MKC genomes available from public databases (Supplementary Data 1), were analyzed for global diversity. In total, we included the genomes of 358 isolates obtained from 18 countries with varying burdens of disease caused by the MKC (Fig. 1a). On average, the MKC genomes are 6.29 Mb in length and contain 5757 protein-coding genes. A genomic alignment of 2280 core genes with at least 90% amino acid identity between strains and covering 2.12 M nucleotides was used to generate an ML phylogeny (Fig. 1b). The phylogeny consisted of seven distinct lineages corresponding to the seven MKC species. The pairwise gANIs within each lineage were all over 98%, while the gANIs between lineages were all below 95% (Fig. 1c), confirming that the lineages should be regarded as different species rather than subtypes[19,20]. Four species (*M. kansasii*, *M. persicum*, *M. pseudokansasii*, and *M. attenuatum*), each containing more than ten strains, were designated as the major species in the current study. A pairwise comparison of single nucleotide variants (SNV) among the strains of each of the four species revealed a median difference of 1888 to

3717 SNVs along the 2.12 Mbp core genome (Supplementary Fig. 1).

**Recombination driving speciation and diversification of the MKC.** Alignment of the 16S, 23S rRNA, and spacer region revealed several sequence mosaics between MKC species (Supplementary Fig. 2), consistent with evolutionary processes in the presence of recombination. A complex evolutionary network was obtained based on 378,876 SNVs along the core genome alignment (Fig. 2a), suggesting that recombination has occurred across large portions of the genome. Analysis of the core-genome alignment with the fastGEAR algorithm identified seven population clusters corresponding to the seven species, with extensive ancestral recombination (occurring during the speciation) and recent recombination (occurring after the speciation) between species that resulted in highly mosaic genomes (Fig. 2b and Supplementary Fig. 3). An average of 411 kb (18.4%) of the core genome was involved in ancestral recombination that contributed to the origin of the species. Recent recombination was detected in all species, with the total genomic fraction of recombinant fragments varying from 0 to 12.2% amongst strains of the different species. For the recent recombination events, most recombinant fragments share high identity (≥99%) with genomic regions of other species, suggesting they represent recombination between species. For the few remaining fragments showing relatively low sequence identity to all of the genomes in our collection, nucleotide identity analysis revealed that they were more similar to sequences of the MKC species than to any other mycobacteria, suggesting that the recombination had occurred with unknown species closely related to the MKC. Removing the SNVs present in the recent recombinant regions significantly decreased the genetic distance between strains for all major species, demonstrating the importance of recombination in the diversification of the MKC species (Supplementary Fig. 1).

The core genome alignment consists of concatenated sequences of discontinuous sequence fragments in each strain, which does not fully represent the features of recombination, i.e., the genomic distribution and length of recombinant fragments. Therefore, recent recombination events were further explored by Gubbins analysis based on whole-genome alignments for each of the four major species. Evidence of recombination was found evenly distributed across the genomes of all four species (Fig. 3b, Supplementary Fig. 4), with fragment lengths ranging from a few base pairs to a maximum of 212.9 kb (Fig. 2c), reminiscent of recombination by distributive conjugative transfer (DCT), a form of horizontal gene transfer in mycobacteria[30]. In each species there were both unique recombinant sequences seen in only one isolate and shared recombinant sequences seen in multiple isolates, demonstrating that recombination events occurred at different stages during the diversification of species. For the clinically associated *M. kansasii*, the recombination donors were mainly from *M. persicum* and *M. pseudokansasii* (Fig. 2b, Fig. 3b). Fragments derived from *M. pseudokansasii* were more common in *M. kansasii* strains from North America, while *M. kansasii* strains from East Asia or Europe had recombined more frequently with *M. persicum* (chi-square test, $p < 1e{-}6$, two-sided).

**Genetic evidence for independent environmental acquisition of clinical infections.** After excluding the SNVs in the recombinant regions, we inferred phylogeny for each of the major MKC species to investigate the genomic differences between isolates based solely on non-recombinant mutations (Fig. 2d, Fig. 3a). Within each species, there were deep branches, where the clinical isolates were separated by thousands of SNVs,

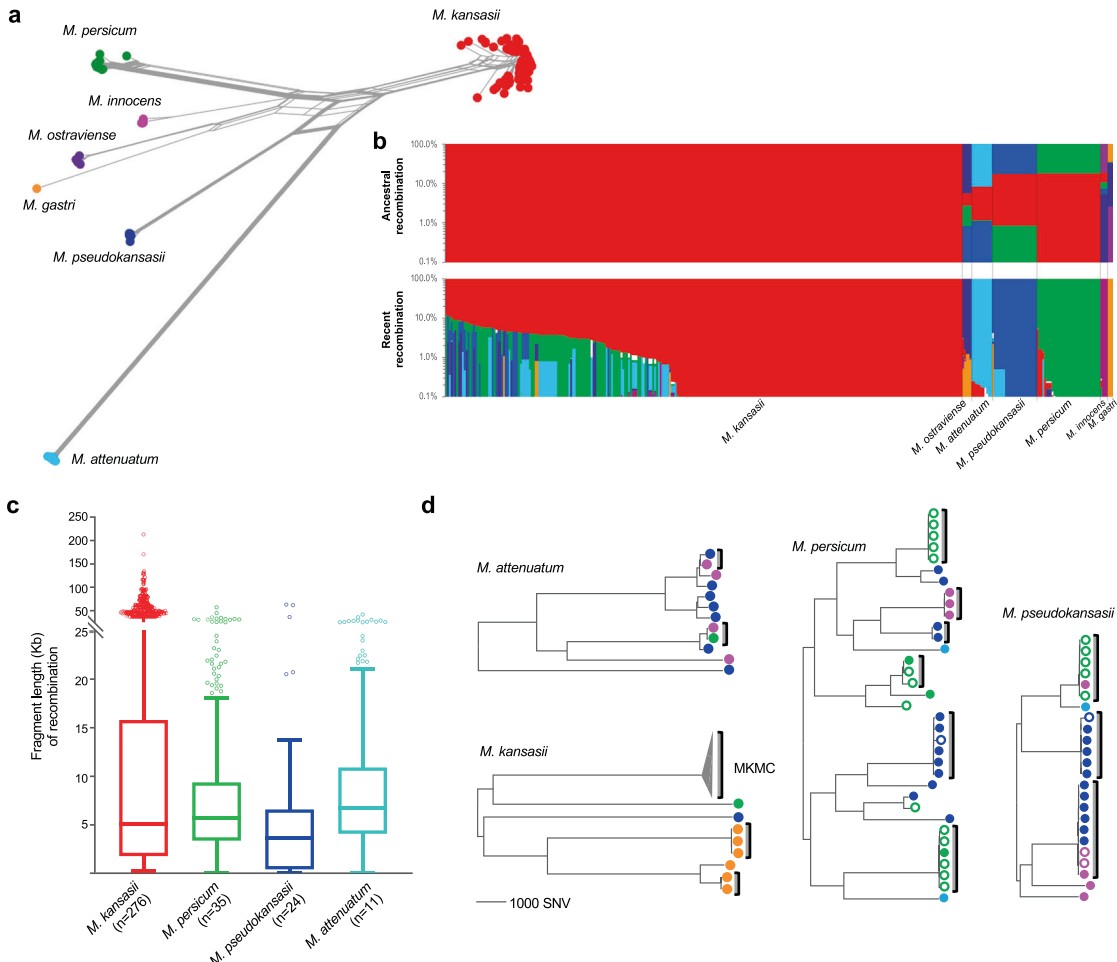

**Fig. 2 Genomic recombination and its contribution in speciation and diversification of the *M. kansasii* complex. a** Phylogenetic network of the *M. kansasii* complex based on the core genome alignment of 358 isolates. **b** Population structure and genomic recombinations inferred by fastGEAR. Each line represents the genomic constitution (exhibited as color strips) of individual isolates according to ancestral (upper panel) or recent (lower panel) recombinations. Strip colors represent the different species as in panel (**a**). White strips (lower section) represent recent recombinations from unknown sources. Source data are provided as a Source Data file. **c** Length distribution of recombinant fragments in the four major species. Boxes show the median and interquartile range (IQR) while whiskers extend to a maximum of 1.5× IQR. **d** Maximum likelihood phylogeny of the four major species based on non-recombinant SNVs. Brackets indicate clusters containing isolates with an average pair-wise genomic difference of fewer than 100 SNVs. The colors of terminal nodes indicate the geographical origins of the isolates, corresponding to Fig. 1a. Filled circles indicate a human source; empty circles, an environmental source. MKMC *M. kansasii* main cluster.

consistent with independent infections caused by unrelated environmental isolates. In addition, there were 14 clusters, covering strains in all four major species, which contained closely related isolates with an average pair-wise difference of fewer than 100 SNVs, suggesting potential dissemination of successful clones. Seven of these clusters contained isolates obtained from both water and human patients, consistent with the environmental strains as the source for the clinical infections. Nine of the clusters contained isolates from a single geographic region, while the remaining five clusters contained strains isolated on different continents.

The largest cluster, with 268 isolates of *M. kansasii*, was named the *M. kansasii* main cluster (MKMC). It contained 79.2% (244/308) of all the clinical isolates included in the current study. The MKMC contained 20 strains isolated from water sources, including one strain from the Czech Republic that clustered with clinical strains from neighboring Poland. The remaining 19 strains were isolated from an exposed cooling tower linked to a geothermal water source in Portland (a small town in southeast Australia) during an outbreak of *M. kansasii* infections in the

1990s. In the maximum likelihood (ML) phylogeny, these water isolates clustered with eight strains isolated from patients in the town, including six patients who were part of the outbreak (Fig. 3a). The phylogenetic (median-joining) network of these 27 isolates from Portland formed a star-burst structure with most descendant strains surrounding the central ancestral genotype (Fig. 3c). Isolates with the ancestral genotype were consistently cultured from water samples between 1990 and 1996, strongly suggesting the cooling tower as the source for this outbreak. The clinical isolates all had different genotypes, and all except one were closest to the ancestral type with genomic differences of 0–7 SNVs. The genotype of the remaining isolate was identical to a water isolate and differed by three SNVs from the ancestral type. This strongly suggests that the human infections were each acquired independently from the contaminated water supply rather than by human-to-human transmission. Although the outbreak had ended by 1996, after the cooling tower was bypassed and the use of geothermal water discontinued, two strains belonging to the same "Portland" clone were isolated from patients in 2005 and 2007 (Fig. 3c).

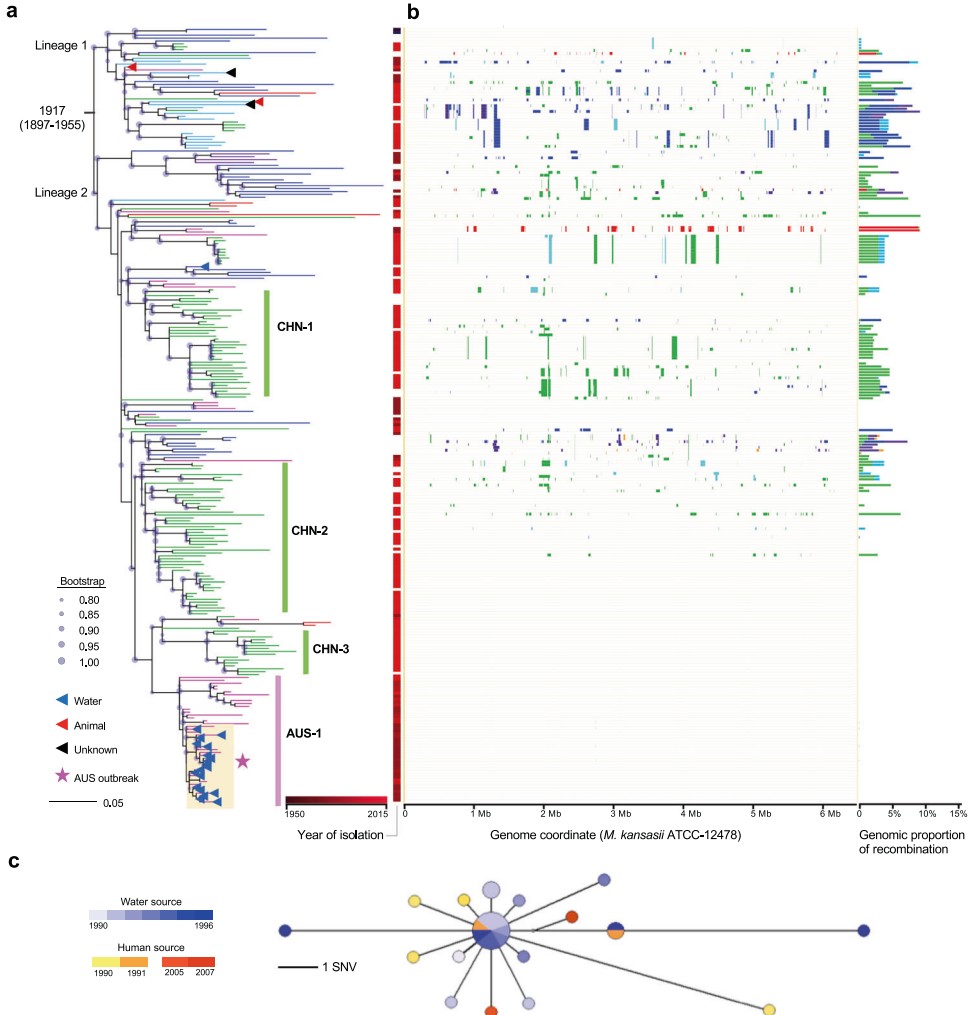

**Fig. 3 Phylogenomic analyses of the *M. kansasii* main cluster (MKMC). a** The maximum-likelihood phylogeny of the MKMC based on non-recombinant mutations. The colors of terminal branches indicate the geographical origins of the isolates, as in Fig. 1a. Isolation from non-human or unknown sources is indicated by triangles in the terminal nodes. **b** Genomic pattern and proportion of recent recombinations for individual isolates. Donor species are colored as in Fig. 2a. Source data are provided as a Source Data file. **c** Median-joining network for the Australia outbreak strain cluster. Node size indicates the number of isolates; node color indicates the source and year of isolation.

**The origin and global dissemination of the *M. kansasii* main cluster**. Phylogeographic analysis of the MKMC revealed two basal branches, designated Lineage 1 and Lineage 2. The strains from China and Australia (121/131 and 57/58, respectively) predominantly belong to Lineage 2, while nearly all strains from USA and Canada (20/21) belong to Lineage 1. Strains isolated in Europe were the most diverse, constituting several branches in both lineages (Fig. 3a), and Bayesian phylogeographic analysis suggested Europe as the most likely origin of the entire complex (Supplementary Fig. 5). Several local branches contained isolates exclusively from China (CHN-1, 2, and 3) or Australia (AUS-1), suggesting early introductions and subsequent local expansions.

We calculated a median Tajima's *D* of −2.45 and −0.93 for individual core genes of the MKMC based on all SNVs or nonrecombinant SNVs respectively (Supplementary Fig. 6), suggesting recent population expansion and/or a potential selective sweep. The isolation time of the *M. kansasii* strains ranged from the 1990s to 2010s, which made it possible to estimate the date of origin of the MKMC and its substitution rate using Bayesian evolutionary analyses calibrated by the sampling dates. This analysis employed a subset of 121 strains with short sequence reads, unambiguous collection dates, and a genomic

recombination proportion less than 1.0%, together with the reference strain ATCC 12478 isolated in 1953[31,32]. The existence of a significant temporal signal was confirmed by both the root-to-tip regression and the date randomization test (Supplementary Fig. 7). The bayesian phylogenetic analysis estimated the date of the MRCA of the MKMC to be around 1917 (1897–1955) (Fig. 3a, Supplementary Fig. 8a) with an evolutionary rate of $1.12e-7$ (95% CI, $7.93e-8$, $1.62e-7$) nucleotide changes per site per year (Supplementary Fig. 8b).

**Genes potentially contributing to the success of *M. kansasii***. The recent expansion of *M. kansasii* and its association with clinical infections suggest that it may have evolved greater pathogenicity for human hosts than the other MKC species. By comparative genomic analysis, we identified 147 genes specific to *M. kansasii*, several of which have been associated with metabolic adaptation or virulence within human hosts (Supplementary Data 2). Among these are three clustered genes encoding enzymes PrpC and PrpD and regulator PrpR, which are components of the methylcitrate cycle (MCC) that eliminates the toxic propionyl-CoA produced during in vivo catabolism of cholesterol and fatty

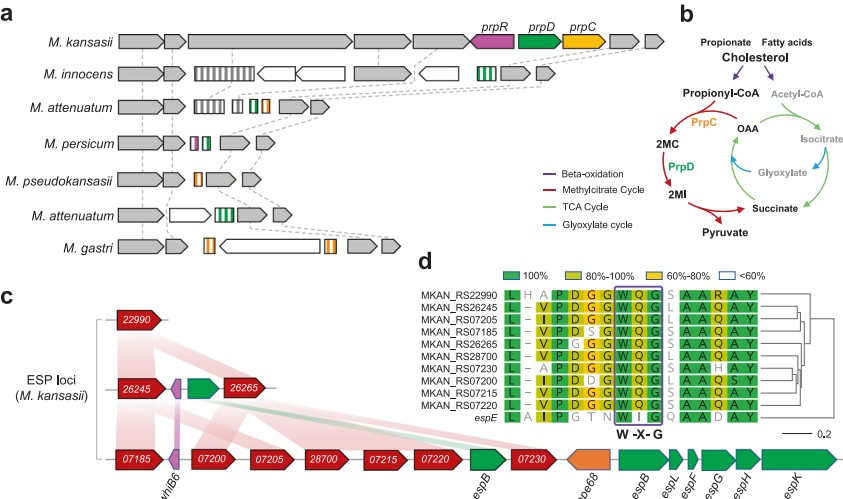

**Fig. 4 Genomic loci specific to *M. kansasii*. a** Synteny map for the genomic region flanking the genes of methylcitrate cycle (MCC) in *M. kansasii*. Full-length and truncated genes are represented by arrows and rectangles respectively. The full MCC genes in *M. kansasii* and their orthologous sequences in the other MKC species are indicated with different colors. The flanking genes are in gray or white to represent homologous or orphan genes, respectively. **b** A scheme of the MCC of mycobacteria and its relation to the beta-oxidation, tricarboxylic acid (TCA) and glyoxylate cycles. 2MC 2-methylcitrate, 2MI 2-methylisocitrate, OAA oxaloacetate. **c** Synteny map of the three ESP (ESX-1 secretory protein) loci specific to *M. kansasii*. Red arrows represent *espE*-like genes and the numbers indicate their MKAN_RS identifiers. **d** Sequence similarity between the EspE of *M. tuberculosis* and the EspE-like proteins of *M. kansasii*. Residues are colored to indicate similarities.

acids (Fig. 4a, b)[33,34]. The MCC genes are located in a highly variable genomic region, and these three, along with a few flanking genes, are completely or partially deleted in the other MKC species. Eighteen of the other *M. kansasii* specific genes encode potential secretory proteins of the ESX-1 system (ESP), a type VII secretion system associated with virulence in *M. tuberculosis*[35]. The genes are distributed in three genomic loci, one of which is comprised almost entirely of ESPs, the WhiB6 regulator, and a PPE protein associated with ESX-1 (Fig. 4c, Supplementary Fig. 9a). Among the 18 ESPs, 10 are paralogs encoding EspE-like proteins and all contain the WxG motif, a characteristic of ESX-1 substrates (Fig. 4d)[36]. A BLAST search revealed that these *espE*-like genes are not present in the other MKC species, nor in most other mycobacteria except *Mycobacterium marinum* and closely related species such as *Mycobacterium ulcerans* and *Mycobacterium liflandii* (Supplementary Fig. 9b). In *M. marinum*, the *espE*-like genes are arranged in tandem immediately upstream of the ESX-1 locus, while in *M. kansasii* the three *espE*-like containing loci are each separated from the ESX-1 locus. Evolutionary analysis suggested that the *espE*-like genes were independently acquired by *M. marinum* and *M. kansasii* (Supplementary Fig. 9c), and then expanded in parallel (Supplementary Fig. 10).

**Genes under positive selection in the *M. kansasii* main cluster.** The fixed non-recombinant mutations (allele frequency ≥95%) of all the MKMC isolates were used to identify convergent mutations or genes containing an unusually high number of mutations, which could be evidence of positive selection[37,38]. Under the neutral model, the number of mutations per gene is expected to follow a Poisson process that predicts a mean of 1.73 genes with four mutations and means proximate to zero genes with five or more mutations. However, we detected 10 genes with four mutations, a 4.76-fold deviation from the Poisson prediction in a neutral model, and 9 genes with five or more mutations (Supplementary Fig. 11). In addition, four genes were found to harbor convergent mutations that had evolved independently at least three times. These 23 genes encode proteins associated with diverse functions, including secondary metabolism (seven genes),

DNA replication, recombination and repair (four genes), metal ion transport and metabolism (three genes) and carbon metabolism (three genes) (Fig. 5a, Supplementary Data 3).

The most polymorphic locus encodes Zur, the regulator of zinc uptake. A total of 36 fixed *zur* mutations were identified in 38 clinical isolates, all of which were nonsynonymous or frameshifts (Fig. 5b), implying loss of function. All *zur* mutations could be mapped to terminal nodes in the phylogeny (Supplementary Fig. 12), and none were found in isolates from aquatic sources, suggesting they could be adaptive mutations that emerged within the human host. Besides the fixed mutations, we also identified 41 unfixed *zur* mutations (allele frequency < 95%) in an additional 35 clinical isolates, with several isolates carrying multiple unlinked mutations (Supplementary Fig. 13), strongly supporting their emergence within the host. The second most polymorphic locus encodes a pair of TetR family transcriptional regulators (TetR1/2) that are potentially involved in Gamma-butyrolactone (GBL) signaling[39]. A total of 15 mutations in these two genes were found in 63 isolates. This locus also exhibited the highest recombination density (Fig. 5a, b), with evidence of 42 independent recombinant events affecting 70 isolates, of which 14 harbored nonsense/frameshift mutations. As opposed to the *zur* mutations, many of the mutations and recombination events in the TetR1/2 locus could be mapped to inner nodes in the phylogeny (Supplementary Fig. 14), suggesting that they may represent early adaptations prior to the human infections.

The third most polymorphic locus contained two genes encoding a putative polyketide synthase (Pks5) and a glucosyltransferase (WbbL2), enzymes that are involved in lipooligosaccharide (LOS) synthesis[40–42]. A total of 14 fixed mutations distributed in 16 clinical strains were detected in these two genes. Additional nonsynonymous mutations, including nonsense and frameshift mutations, were detected in neighboring genes encoding other enzymes associated with LOS synthesis: a fatty acyl-CoA synthetase (*fadD25*), an acyltransferase (*papA3*), and two glucosyltransferases (*gtf1* and *gtf2*) (Fig. 5c). All the fixed mutations were exclusively found in clinical isolates and all but one of them could be mapped to terminal nodes in the phylogeny (Supplementary Fig. 12). Furthermore, unfixed mutations in

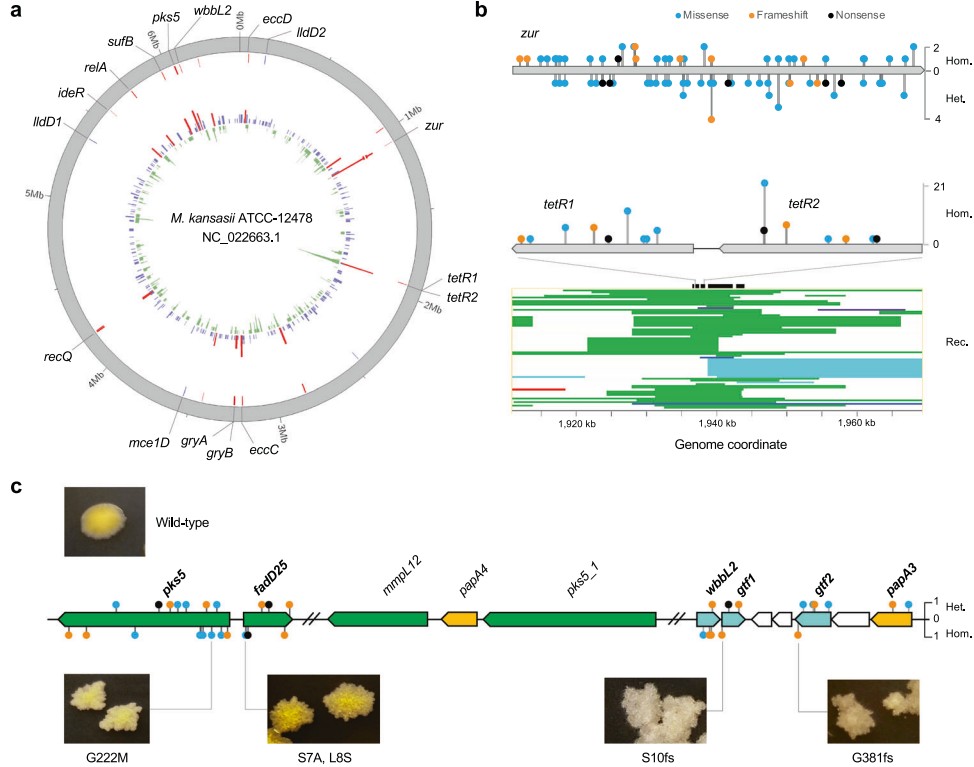

**Fig. 5 Genes under positive selections in the MKMC. a** Circular plot of genes under potential positive selection. Innermost inward bars, number of recombination per gene; innermost outward bars, number of mutations per gene (red bars, *n* ≥ 4); outer red lines, location of highly polymorphic genes; outer blue line, location of genes with convergent mutations. **b** Schematic diagrams depicting the distribution and frequency of non-synonymous mutations in *zur* and *tetR1/2*, and the recombinations around *tetR1/2*. Hom. homozygous mutation, Het. heterozygous mutation, Rec. recombination (strip colors represent donor subspecies corresponding to Fig. 2a). Source data are provided as a Source Data file. **c** Schematic representation of mutations in LOS biosynthesis genes and corresponding morphology of the mutant strains. Genes were colored according to their functions. Green, genes involved in polyketide synthesis; orange, acyltransferase; cyan, glucosyltransferase. fs frameshift. Source data are provided as a Source Data file.

those genes were identified in 18 clinical isolates (Supplementary Data 3), suggesting they likely represent adaptive mutations emerging within the host. Loss-of-function mutations in LOS biosynthesis genes have been associated with the transformation from smooth to rough colony morphology in other mycobacteria[42,43]. Colony morphology was described for 110 of the clinical strains from Shanghai (Supplementary Table 2, Supplementary Data 1), of which 12 were recorded as having rough colonies. Of these 12 rough colony strains, all harbored mutations in the putative LOS synthesis genes, while none were found in the 98 smooth colony strains. This high correlation between the mutations and morphology suggests that these mutations affect LOS synthesis.

Convergent mutations that evolved at least three times were identified in the regions upstream of three genes encoding two L-lactate dehydrogenases (*lldD1*, −12 C > T in three isolates; *lldD2*, −44 C > T in six isolates) and a lipase (MKAN_RS10545, −157 G > A in four isolates), which are involved in lactate and lipid metabolism, respectively. The mutations were exclusively identified in clinical isolates, and an additional 14 clinical isolates were found to harbor fixed or unfixed mutations in these regions (Supplementary Data 3). Convergent mutations were also found in the coding region of *mce1D*, a subunit of the Mce family transporter that is putatively responsible for lipid/sterol transportation[44]. Two different nucleotide substitutions (970 G > C/T) were identified in codon 324 of *mce1D*, both resulting in the same amino acid substitution, suggesting they were likely gain-of-function mutations. Among the 12 isolates with a recombinant *mce1D*, five harbored the 970 G > C mutation.

## Discussion

The population genomic analyses of global *M. kansasii* yielded several insights into the population diversity, epidemiology, evolution, and host adaptation of this important pathogen. The phylogenomic analysis confirmed that previously defined *M. kansasii* subtypes represent closely related species, as has been recently proposed[19,20]. We found ample evidence of ongoing homologous recombination between the species, but no trace of recombination with any other mycobacterium species, further supporting the classification of these closely related species as the MKC[20]. Extensive recent and ancestral recombination events, likely driven by DCT, resulted in the mosaic genomes observed in the MKC species[30], thereby demonstrating the importance of DCT in both the speciation and diversification of these species. Notably, there was evidence of recent recombination in the 16S–23S rRNA locus (Supplementary Fig. 3), emphasizing that species/subspecies identification should include multiple genes.

The comparative genomic analysis revealed genes associated with metabolism and virulence that were found only in the predominant *M. kansasii* strains, and perhaps contribute to their success in colonizing and causing disease in humans. When mycobacteria infect humans, they use host cholesterol and fatty acids as carbon sources, but the beta-oxidation of cholesterol and odd-chain fatty acids generate propionyl-CoA that is toxic to the bacilli[33,34,45,46]. Pathogenic mycobacteria alleviate the toxicity by consuming propionyl-CoA through the MCC cycle, which is important for the growth of mycobacteria within macrophages[34,46]. The maintenance of the MCC genes in *M. kansasii*, and their absence in the other MKC species, may

represent an adaptation to the host that partially explains why this species is most often associated with clinical infections. The analysis also found that *M. kansasii* contains a unique cluster of *espE*-like genes encoding potential ESX-1 substrates resembling EspE, a highly abundant mycobacterium cell surface protein secreted through the ESX-1 system[47]. A similar *espE*-like gene cluster is present in the genomes of *M. marinum* and closely related species, and their inactivation led to defective granuloma formation in zebrafish embryos[48]. It is therefore possible that the *espE*-like genes in *M. kansasii* also encode similar secretory proteins involved in modulating the host immune response and causing disease, although further confirmations by in vitro and in vivo studies are needed. An *M. kansasii* strain was recently isolated from a river fish with granulomatous nodules[49], and the reference strain ATCC 12478 can cause chronic infection and granulomas in zebrafish embryos[50], suggesting that fish may be a host for *M. kansasii*. Since *M. marinum* is also known to cause infections in fish, a common host might explain the parallel evolution of the *espE*-like gene clusters in the two species. A more recent study revealed that an *espACD* operon that is exclusively present in *M. kansasii* may be associated with its pathogenicity[51]. We identified four highly divergent *espA* paralogs with diverse distributions in the MKC species and each of them is part of a putative *espACD* operon (Supplementary Fig. 15). In *M. kansasii*, there are three *espA* paralogs, one of which (MKAN_RS22010) is unique to *M. kansasii* and the other two (MKAN_RS11540 and MKAN_RS12085) including the ortholog of *M. tuberculosis espA* (MKAN_RS12085) are both present in some other MKC species. The fourth paralog is present in all MKC species except *M. kansasii*. Given the evolutionary complexity, the contribution of *espA* paralogs to the pathogenicity of *M. kansasii* is worthy to be further investigated.

Isolates of *M. kansasii* predominantly belong to a homogenous cluster designated MKMC, and its clinical predominance but rare isolation from city water sources raises the possibility of human-to-human transmission. However, by investigating an outbreak of *M. kansasii* infections in Australia, we found genetic evidence that the patients were more likely to have acquired their infections independently from *M. kansasii* strains present in the city water system. This is consistent with previous suggestions that city water distribution systems constitute the principal reservoir for *M. kansasii*[5,52]. Our evolutionary analysis estimated that the MKMC originated in the early 1900s, possibly in Europe, although both the proposed date and geographic origin need to be confirmed by sequencing additional isolates from diverse global regions, especially in America and Africa. Considering the association of *M. kansasii* infections with urban areas, we speculate that the initial expansion of the MKMC was associated with the rapid urbanization of Europe since the 1900s[53] and that it then spread and expanded with the urbanization of other global regions. Although it is unclear how the water-born MKMC could have achieved global dissemination during the past century, systems for storing potable water during long voyages could have played a role.

Several genes involved in metabolism and the stringent response appear to be under positive selections in the MKMC. Host cell lipids are a major carbon source for mycobacteria during infection and are critical for the survival of bacteria within the host[54]. We observed convergent mutations in a subunit (Mce1D) of a putative lipid/sterol transporter and also in the upstream region of a putative lipase involved in the lipid hydrolysis[44], both of which may represent adaptations to a lipid-rich environment within the host. Besides lipids, host cell lactate was recently revealed as an important carbon source for bacterial growth within human macrophages[55]. More recently, convergent mutations in promoter and coding regions of lactate

dehydrogenase gene *lldD2* were extensively identified in *M. tuberculosis*. These mutations were thought to represent an adaptation to changes in host ecology and were associated with higher transmissibility[56]. We also identified mutations upstream of *lldD1/2* in 14 MKMC clinical isolates, emphasizing the importance of lactate metabolism in host adaptation of mycobacteria. The convergent mutations in the regions upstream of *lldD1/2* and the lipase gene likely resulted in upregulation of the corresponding enzymes, which could enhance the metabolic capabilities of *M. kansasii* within the host and facilitate its survival and replication.

Potential positive selection was also identified in genes involved in metal ion acquisition, which may represent an adaptation to the limitation of metal ions (i.e., nutritional immunity) within the host[57,58]. The highest polymorphism was found in *zur*, which in *M. tuberculosis* encodes a transcriptional repressor that regulates the expression of genes involved in zinc and iron uptake and zinc mobilization[59,60]. Inactivation of Zur could increase the expression of genes that improve the ability of *M. kansasii* to compete with the host for the acquisition of zinc and iron (Supplementary Fig. 16). Two additional genes that encode the iron-dependent repressor (IdeR) and the subunit B of the SUF system (SufB) for Fe–S cluster biosynthesis also showed evidence of potential positive selection. Mutations in these genes may augment the ability of the bacilli to maintain iron homeostasis in the iron-limited environment encountered during infection[58,61] (Supplementary Fig. 16).

Seven genes associated with secondary metabolism showed potential positive selection, with nonsense and/or frameshift mutations likely causing loss-of-function. Among these are two TetR family regulators of GBL signaling and two genes involved in LOS biosynthesis. GBL signaling molecules are involved in the regulation of secondary metabolism and morphological development in actinomycetes[39]. The most-studied GBL signaling is the A-factor system associated with secondary metabolism and sporulation in Streptomyces[62,63]. Many of the mutation and recombination events in these two genes mapped to inner nodes of the ML phylogeny, suggesting that they could have been acquired before infecting humans, probably in city water distribution systems. While there is no solid evidence of mycobacterium sporulation, inactivation of these regulators could nevertheless modulate bacterial metabolism to facilitate the survival of *M. kansasii* in the urban water systems, where they would encounter low levels of nutrients and disinfectant residuals. LOS are polar glycolipids associated with cell wall hydrophilicity in several mycobacteria[27,42]. We found a high correlation between loss-of-function mutations in these genes and rough colony morphology in clinical isolates, consistent with previous findings in other mycobacteria[42]. The rough phenotype resulting from the absence of LOS has been associated with enhanced within-host survival and increased virulence in several mycobacteria, including *M. kansasii* and *M. marinum*[29,64]. In *M. tuberculosis*, a recent study demonstrated that the loss of LOS occurred in its *Mycobacterium canettii*-like ancestor and may have played a vital role in its evolution from an environmental mycobacterium to a professional pathogen[43]. Similarly, the mutations in the LOS synthesis genes of *M. kansasii* could also represent in-host selection for increased virulence and the ability to establish a persistent infection within the host.

The extensive selection of mutations in clinical MKMC isolates represents a feature of opportunistic infections, where adaptive mutations are rapidly selected due to the shift from the environment to host niches[65]. Several mutations, including those in the LOS synthesis genes and the *lldD2* promoter of the MKMC strains, mimic the ancestral adaptations of *M. tuberculosis* to a human host and suggest that *M. kansasii* may have the potential

to evolve into a professional pathogen. The putative adaptive mutations we identified in the MKMC isolates all mapped to terminal phylogenetic nodes, suggesting that these putatively more human-adapted strains were not transmitted. However, considering the high similarity of pulmonary infections caused by *M. kansasii* and *M. tuberculosis*, we cannot exclude the possibility of aerosol transmission of *M. kansasii*, which would provide opportunities for multiple rounds of host adaptation. This should raise concern that some members of this species, particularly in the MKMC, may have the potential to evolve into highly adapted human pathogens.

## Methods

**Strain collection and whole-genome sequencing.** In China, patients with suspected pulmonary tuberculosis are referred to local designated hospitals. Positive cultures of putative NTM are subjected to species identification by 16S rRNA PCR sequencing using the forward primer 16S-P1 (TGGA-GAGTTTGATCCTGGCTCAG) and reverse primer16S-P2 (ACCGCGGCTGCTGGCAC) in these hospitals or by a local or national branch of the Center for Disease Control and Prevention (CDC). Clinical isolates collected by the China CDC ($N = 22$, from the national survey of drug-resistance during 2007–2008), Shanghai CDC ($N = 110$, collected 2009–2013), and The Third People's Hospital of Shenzhen ($N = 5$, date of collection unknown) were included. Water isolates ($N = 15$) were obtained from public tap water across Shanghai in 2015, using a filtration method[66]. Briefly, 1 l of water was passed through a membrane filter (pore size, 0.22 μm; Millipore). The membrane was decontaminated by 15 ml of 3% sodium dodecyl sulfate and 1% NaOH for 30 min. The solution was neutralized with 40% phosphoric acid solution and then centrifuged for 15 min at $2000 \times g$. The sediment was then resuspended in about 500 μl of the supernatant and plated on 7H10 plates. In Canada, clinical isolates ($N = 14$, collected 2007–2010) were obtained from the McGill University Health Centre mycobacteriology laboratory and identified as *M. kansasii* by the Laboratoire de Sante Publique du Quebec by 16S rRNA PCR and DNA sequencing. In Australia, clinical isolates ($N = 74$, collected 1990–2015) were referred to the mycobacterial reference laboratory at the Victorian Infectious Diseases Reference Laboratory (VIDRL) and identified by 16S rRNA PCR DNA sequencing. An addition of 28 clinical isolates, collected in 1990–1992 from global areas (Switzerland, $N = 6$; Belgium, $N = 5$; South Africa, $N = 5$; the USA, $N = 5$; the UK, $N = 4$; Japan, $N = 3$) and stored in VIDRL, were also included[67]. For clinical samples, specimens were cultured on Löwenstein Jensen (L–J) slants, and multiple colonies that grew on the slants were scraped for DNA extraction. For water samples, specimens were primarily plated on 7H10 plates, from which a single colony was picked and sub-cultured on L–J slants. Multiple colonies that grew on the slants were scraped for DNA extraction. Genomic DNA was sequenced on either an Illumina Hiseq 2000 or NextSeq 500 platform in single or paired-end mode with an expected depth of 100. Publicly available genomic sequences and short-read data were downloaded from the Assembly and SRA databases of NCBI respectively (Supplementary Data 1).

**Genome assembly and genomic nucleotide identity analysis.** Public sequencing data were downloaded from NCBI and then converted into fastq files using the NCBI SRAtoolkit (2.10.8, https://ncbi.github.io/sra-tools/). Sequencing reads were trimmed and filtered using Trimmomatic (v0.30)[68] and draft genomes were assembled using SPAdes (v3.13.1)[69] in the carful mode with reading correction, auto-sized k-mers, and mismatch corrections. The quality of assembly was evaluated using Quast (v5.02)[70] and contigs of less than 200 bp were filtered out. Pairwise genomic ANIs were calculated using fastANI (v1.1)[71] with default parameters based on the assemblies.

**Core genome analysis.** Core genes for all MKC isolate included in this study were analyzed by Roary (v3.11.2)[72], through which draft genomes were first annotated using Prokka (v1.14.6)[73], and then homologous genes were clustered using the CD-Hit and MCL algorithms. To generate the core-genome alignment, the parameters were set to a minimum of 90% blastp identity, 100% coverage (i.e., the gene must be present in all isolates), and no paralog splitting (i.e., clusters containing paralogous genes were filtered out). Sequences of individual core genes were aligned with MAFFT (v7.407)[74] and then concatenated into a core genome alignment according to their genomic coordinates in the reference genome (NC_022663.1). RaxML (v8.2.12)[75] was used to construct the ML phylogeny based on the core genome alignment with a GTR model and 1000 rapid bootstrap replications. iTOL (v5.7)[76] was used for displaying and annotating phylogenies. SplitsTree (v4.14.5)[77] was used to construct the phylogenetic network by the NeighborNet algorithm based on the core genome alignment. The Tajima's D statistic was calculated for individual core genes of the MKMC using PopGenome (v2.7.5)[78] based on all and non-recombinant SNVs, respectively. For identification of *M. kansasii* specific genes, a minimum of 80% blastp identity with paralog splitting was set in Roary.

Genes present in all *M. kansasii* isolates but absent in all isolates of any other MKC species were selected.

**Population structure and recombination analysis.** Population structure was inferred using hierBASP[79]. The core genome alignment was subjected to hierBAPS analysis with a uniform prior on the number of clusters. Genomic recombination was inferred using fastGEAR[80] based on the core genome alignment with an integration number of 15 (default value). The fastGEAR used BAPS to define the "best" number of clusters and then detect "lineages" that are genetically distinct in at least 50% of the alignment. fastGEAR detects both ancestral recombinations that affect all isolates in a lineage as well as recent recombination that affects a subset of isolates in a lineage. For each ancestral recombination, the larger lineage was assumed to be the donor (by default).

The recent recombinations in isolates of the major species (*M. kansasii*, *M. persicum*, *M. pseudokansasii*, and *M. attenuatum*) were further explored with Gubbins (v2.3.4)[81]. The whole-genome alignment of each species was generated by an in-house pipeline based on Minimap2 (v2.17)[82]. Briefly, the contigs of individual strains were aligned to the reference genome by Minimap2 with the preset parameter "asm20" for the alignment. By filtering secondary and short alignments (<200 bp), the nucleotide corresponding to the reference genome at each site is determined to generate the "pseudogenome" for each isolate. Pseudogenomes of each species were concatenated to make a whole-genome alignment, which was subjected to Gubbins for recombination identification with default parameters. The donors of the recombination fragments were determined by a BLAST (v2.9.0)[83] search against a local database containing all of the MKC genomes included in the current study and the representative reference genomes for other mycobacteria collected in the NCBI database. A cutoff value of identity was set to 99% to identify the probable donor strains. The outputs from Gubbins were viewed with Phandango (v1.1.0)[84], or by an in-house python script that plots the recombinations with information including genomic coordinates and donor species.

**Mapping based analysis.** We applied mapping-based analysis to study the genetic variants among the MKMC strains. The trimmed reads were mapped to the reference genome ATCC 12478 by Bowtie2 (2.3.5)[85] and variants including SNVs and short-indels were called by a SAMtools (v1.9)/VarScan (v1.4.3) pipeline[86,87]. Variants were called at loci where the alternative base calls were supported by at least five reads that aligned to the reference in both forward and reverse directions. Variants in repeat regions, putative PE/PPE family genes, and transposable elements were excluded. Variants supported by ≥95% of the mapped reads were defined as fixed/homozygous mutations, otherwise, variants were defined as unfixed/heterozygous mutations. The homozygous SNVs in non-recombinant regions detected by both mapping- and assembly-based analysis were used to construct the ML phylogeny of the MKMC by RaxML based on the GTR model. We found several extraordinarily long terminal branches in the ML phylogeny for isolates from PRJ374853, which likely represent assembly errors, and the corresponding terminal branches were thus truncated to 0. Network (v5.0)[88] was used to generate median-joining networks for the outbreak strains from Australia based on the concatenated SNV sequences.

**Bayesian evolutionary analysis.** The geographic origin of the MKMC was analyzed using the Bayesian Binary MCMC (BBM) method integrated into RASP (v4.0)[89]. The BBM method inputs the posterior distribution of Bayesian inference to reconstruct the possible ancestral distributions of given nodes via a hierarchical Bayesian approach. The ML phylogeny of the MKMC constructed in the above section was used for the analysis. The strains were classified into six geographic regions based on where they had been isolated: East Asia, Australia, Europe, North America, South America, or South Africa. The Bayesian analysis was run with a fixed JC model for 5,000,000 cycles, 10 chains, a temperature parameter of 0.1, with sampling every 100 generations. Bayesian dating of the phylogeny was based on a subset of 121 strains of the MKMC with short-read data (to exclude assembly errors in publicly available genomes), clear dates of isolation, and genomes with proportions of recombinations <1.0%. The reference strain ATCC 12478, which was isolated in 1953, was also included. The temporal signal in the sequence alignments was investigated using TempEst (v1.5.3)[90]. As a complementary assessment of the temporal signal in the data, a date randomization test was performed on our datasets with the R package TipDatingBeast (v1.1-0)[91]. Sampling dates of the strains were randomly shuffled 20 times, and the randomized datasets were analyzed with BEAST (v1.8.0)[92] using the same parameters as for the original datasets. If the 95% HPD intervals of root-height obtained from the original data do not overlap with the estimates obtained from the randomized datasets, a statistically significant temporal structure could be confirmed. BEAST was used to determine the timescale and the evolutionary rate of the MKMC using the tip-date calibration based on the whole-genome alignment. We used an uncorrelated log-normal distribution for the substitution rate and constant population size for the tree priors. The analysis was done in three chains of $5 \times 10^7$ generations sampled every 1,000 generations to assure independent convergence of the chains. Convergence was assessed using Tracer (v1.6)[93] to ensure that all relevant parameters reached an effective population size of >100.

**Detection of genes under positive selection in the MKMC**. Genes with convergent mutations or high numbers of non-recombinant mutations could have been subject to positive selection. A subset of 247 MKMC isolates with short reads data was used for the analysis. To identify genes with multi-diverse signatures, the non-recombinant homozygous mutations of all 247 MKMC isolates were used to calculate the mutation density (number of mutations per gene). Under the neutral evolution model, the number of substitutions per gene is expected to follow a Poisson process. The 95% confidence interval of mean predicted values from a Poisson distribution was estimated based on the Wald interval for the mean, and a significant deviation from the interval was taken as a signal of potential positive selection[37,38]. To identify convergent mutations, non-recombinant homozygous mutations were analyzed against the maximum-likelihood phylogeny by TimeTree (v0.6.4)[94]. Homoplastic mutations independently evolved at least three times were identified as under potential positive selection. A circular plot was created using ClicO FS (v1.0)[95] to display gene loci, recombination, and mutation densities of individual genes.

**Reporting summary**. Further information on research design is available in the Nature Research Reporting Summary linked to this article.

## Data availability

Sequence data associated with this study were deposited in the Sequence Read Archive (SRA) of NCBI under project accession PRJNA323639. Accessions for publicly available genomic data are given in Supplementary Data 1. Source data are provided with this paper.

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

## Acknowledgements

We thank Aina Sievers and Kathy Jackson for their expert technical support. This work was supported by the Natural Science Foundation of China (81661128043 and 81871625 to Q.G., 81902107 to Q.M.), National Science and Technology Major Project of China (2017ZX10201302 and 2018ZX10715012 to Q.G., 2018ZX10103001 to Y.Z.), National Key Research and Development Program of China (No. 2017YFD0500301 to J.Y.), Sanming Project of Medicine in Shenzhen (SZSM201611030 to Q.G.), Science and Technology Department of Sichuan Province (2018JY0135 to T.L.), Non-coding RNA and Drug Discovery Key Laboratory of Sichuan Province (FB19-02 to T.L.), National Health and Medical Research Council of Australia (GNT1105525 to T.P.S.), CIHR Foundation Grant (FDN-148362 to M.B.), Guangdong Provincial Science and Technology Program (No. 2019B030301009 to X.C.), Shenzhen Science and Technology Project (JCYJ20170412151620658, JCYJ20170307095303424 to X.C.)

## Author contributions

Q.G., T.L., T.P.S., M.A.B., Q.P., X.C., and Yanlin Z. designed the study. P.X., Yangyi Z., J.L.P. and M.Gh. processed the samples and extracted DNA. T.L. analyzed the data and prepared figures and tables. T.L., H.E.T., Q.G., T.P.S., M.A.B., Q.L. and M.Gh. interpreted the data and wrote the paper. Y.J., J.L., Q.M., B.H., B.P.H., J.A.M.F., M.Gl., W.H., P.H., Y.W., and H.L. participated in sample collection and preparation.

## Competing interests

The authors declare no competing interests.
