## [Peer Review File · Nature Communications]

REVIEWER COMMENTS

Reviewer #1 (Remarks to the Author):

The paper "Population genomics provides insights in diversity, epidemiology, evolution and pathogenicity of the waterborne pathogen *Mycobacterium kansasii*" by Luo et al is impressive and adds significantly to our understanding of the opportunistic pathogen *M. kansasii*. The paper coherently describes the diversity within the species (and newly coined subspecies) and pinpoints clear indications of human adaptation in clinical isolates. The study employs an impressive range of methods and the results are beautifully presented.

My comments/critique is mostly superficial, and listed below:

1. Line 136. The authors should make it clear the the low median Tajima's D across subspecies 1 is suggestive of population expansion AND/OR a selective sweep.
2. Lines 139-141. The sentence is not coherent: the mosaics dont infer a reticulate network, the authors/software do, based on the mosaicism.
3. Line 143. Replace "across the whole genome" with "across large portions of the genome"?
4. Line 144. Add citation to fastGEAR here?

Line 177 and throughout the text: "de novo" is not an obvious term to use when referring to SNP introduced by mutation rather than recombination. The term is generally used to separate inherited (which could be either the result of mutation or recombination) from novel mutations. The authors should either use another term, or explain in an unequivocal manner what they mean by "de novo" mutations.

Line 189: "Czech" is not the name of the country and should be replaced with Czech republic.

Line 255: I dont think it's correct to state that four mutations in a gene represents a significant deviation from the inferred Poisson distribution? If so, what statistic was used to assess significance?

Figure 3: the country annotations are not easily gleaned from the figure. This is particularly challenging when following the presentation on lines 207 and onwards. Perhaps some colored dots on the terminal branches would be better.

Vegard Eldholm

Reviewer #2 (Remarks to the Author):

In this study, Luo and Xu, et al. sequenced a large collection of *Mycobacterium kansasii* and analyzed the relationship of major clades to previously described subtypes, identified recombination within and between subspecies, examined the relationship between clinical and environmental isolates, performed dating and phylogeographic analyses for a lineage comprising many of the clinical isolates, and identified gene content differences and loci under selection. These analyses substantially contribute to the current literature on *M. kansasii* genomics, and the authors have made the sequencing data publicly available and included detailed metadata in their supplementary tables.

However, I have two major concerns that should be addressed before publication.

1. How does the sampling effect the conclusions of the phylogeography analysis? Specifically, I am concerned that a more diverse sampling effort in Europe is leading to the conclusion of a European ancestor.

2. How does the culturing methodology affect the conclusions about positive selection? The authors primarily examine mutations occurring at terminal nodes or at intermediate frequencies in sequencing data for individual samples. I am not currently convinced that these results are not resulting from in vitro adaptation rather than host adaptation based on the details given in the text.

Detailed comments:

54: Since the conclusion that DCT is the likely mechanism of recombination is based on only the lengths of detected recombinant portions of the genome rather than demonstrating that DCT occurs via mating experiments, this sentence in the abstract needs to express some uncertainty about this conclusion (perhaps, "that may be mediated by...").

132-136: Positive Tajima's D values can also indicate a decrease in population size. Also, the positive values reported here are small (close to zero), so these values might be the result of neutral evolution. The negative value reported could be indicative of population expansion or a selective sweep. Given the amount of recombination reported by the authors, is a selective sweep acting on one locus likely to affect the whole genome?

139-141: The 16S and 23S for phylogenetic network analysis does not add much here since a network based on the whole genome is also reported.

162-167: It would be helpful for readers to know that Gubbins was used to detect recombination in this analysis since the authors used FastGEAR in the previous paragraph, and Gubbins and FastGEAR detect different recombination events (FastGEAR only identifies recombination between clades/clusters).

213: Have the authors considered the effects of sample size on the phylogeographic analysis? Isolates from North America, South America, Africa, Asia (outside of China), and Australia appear to come from 1-2 locations within the continent. Isolates from Europe may be the most diverse because it seems that a geographically diverse sample was obtained based on Figure 1A.

257-258: For mutations that were found to independently evolve, did the authors check to see if these mutations were present in recombinant regions?

264-267: Can the authors provide more detail on the culturing and sequencing protocol for these isolates? Are the cultures likely representative of mutations present in patients or are significant bottlenecks expected? Was culturing done the same way for clinical and environmental isolates? Is it possible that mutations found on the terminal branches or at intermediate frequencies are adaptations to in vitro culturing rather than host adaptation or mapping/sequencing errors?

310: homologous instead of homogenous?

455: hierBAPS is misspelled

Figure 1A.

It would be helpful to know where the underlying data supporting the prevalence of *M. kansasii* depicted in the map come from (sample collection method, year, etc.).

Does the size of the symbol representing geographic origin of the isolates have any meaning?

Figure 1/2/3.

Could you use a different color scheme for the lineages compared to the isolations source in Figure 1?

Figure 3A.

Interestingly, there seems to be some completely clonal clades within lineage 2 in this tree. Is there any evidence that these strains may have lost their ability to participate in DCT (potentially, mutations in the ESX loci)? Or for some reason do not come into contact with other *M. kansasii* strains (potentially human to human transmission)? Or these strains have recently emerged and there hasn't been time for a recombination event to occur (what is the TMRCA for these strains)?

Table S3. Potential is misspelled.

Reviewer #3 (Remarks to the Author):

I read with interest the manuscript by Luo et al on the evolutionary history and global diversity of *M. kansasii*. This is a very well written manuscript incorporating a very relevant and globally diverse sample. The methods are detailed and sound. In this work the authors confirm the different phylogenetic lineages formed by different subtypes along with the global evolutionary history of the *M. kansasii* main complex and for which the authors can identify different lineages with distinct dispersion patterns. The latter finding is novel and important to provide an adequate background to the study of the evolving pathogenic ability by this species. The genes proposed to contribute to the virulence/pathophysiology of this species will require further confirmation by in vitro and in vivo studies. Also, the study should have included more environmental isolates as some settings may lack the adequate environmental context.

Some comments:

The authors state that at least six *M. kansasii* subtypes exist while the existence of seven subtypes is widely recognized based on PCR-RFLP. Please clarify and acknowledge that no strain from subtype VII was included in the study. The authors did gather a large *M. kansasii* genomic dataset but the absence of subtype VII leaves a gap in the study.

The authors start by extrapolating the global diversity of *M. kansasii*, confirm the monophyletic nature of the different subtypes and propose that these should be considered as subspecies rather than subtypes based on genome-wide ANI. The results obtained in this regard are not novel, similar findings have been described by Tagini et al (2019) which in fact propose that the different subtypes of *M. kansasii* should comprise novel species. The authors are not using the currently standing nomenclature, instead propose a new one which in my opinion creates further confusion. The standing nomenclature should be incorporated throughout the manuscript or, alternatively, the authors can explain the reason of not adopting the standing nomenclature based on the evidence emerging from these results. The novel nomenclature proposed by Tagini et al should, nevertheless, be mentioned in the introduction.

A low Tajima's D was observed for ssp. I. This was calculated based on sliding windows throughout the genome, although the authors do not provide an explanation on why not using overlapping windows. I believe it would be interesting to evaluate this metric on an individual gene basis. My main concern here is if this diverging value may be driven by the large sample size for ssp. I when compared with the other subtypes due to sampling bias. The SNV distribution is clearly different in ssp. I.

Lines 176-186 – This part is not easily perceived from the figures that are provided or, the annotation of these is insufficient. The fourteen complexes are not clearly identified.

Regarding the dating analysis, Fig.S7 does not convey the robust temporal signal that is mentioned in Line 211. Please explain. Also, the MRCA date is placed around 1917, the mutation rate is very low and the perspective offered in the discussion is somewhat speculative. What sequences were used as input for BEAST? The core genome or the SNVs obtained by mapping?

In the Methods section, please elaborate further on the collection strategy used for the isolates included in the study along with date intervals for each subset.

Fig.S10 – Please check the spelling of *M. kansasii*.

In the revised manuscript we followed the recommendation of Reviewer #3 to use recently proposed nomenclature for the *M. kansasii* complex (MKC). However, to avoid confusion in the following responses, we will use the nomenclature employed by each reviewer in their comments.

Additionally, we realized that, in our initial submission, it's inappropriate to use the word "complex" to describe a set of closely related strains (average pairwise genomic distance <100 SNVs) of the same species, although no reviewer mentioned it. In the revised manuscript we renamed it as "cluster", e.g., the *M. kansasii* main cluster (MKMC).

Reviewer #1 (Remarks to the Author):

The paper "Population genomics provides insights in diversity, epidemiology, evolution and pathogenicity of the waterborne pathogen *Mycobacterium kansasii*" by Luo et al is impressive and adds significantly to our understanding of the opportunistic pathogen *M. kansasii*. The paper coherently describes the diversity within the species (and newly coined subspecies) and pinpoints clear indications of human adaptation in clinical isolates. The study employs an impressive range of methods and the results are beautifully presented.

My comments/critique is mostly superficial, and listed below:

1. Line 136. The authors should make it clear the low median Tajima's D across subspecies 1 is suggestive of population expansion AND/OR a selective sweep.

Response: We agree and have noted this in lines 213-215.

2. Lines 139-141. The sentence is not coherent: the mosaics don't infer a reticulate network, the authors/software do, based on the mosaicism.

Response: The passage has been altered to reflect this comment in lines 135-137.

3. Line 143. Replace "across the whole genome" with "across large portions of the genome"?

Response: The change has been made in lines 138-139.

4. Line 144. Add citation to fastGEAR here?

Response: Due to the limitation on the number of references permitted, the citations for the softwares are shown as weblinks in the Materials and Methods section. The fastGEAR website is cited in lines 478-479.

Line 177 and throughout the text: "de novo" is not an obvious term to use when referring to SNP introduced by mutation rather than recombination. The term is generally used to separate inherited (which could be either the result of mutation of recombination) from novel mutations. The authors should either use another term, or explain in an unequivocal manner what they mean by "de novo" mutations.

Response: We have replaced the term "de novo" with "non-recombinant" throughout the text.

Line 189: "Czech" is not the name of the country and should be replaced with Czech Republic.

Response: The Czech Republic now appears in line 187.

Line 255: I don't think it's correct to state that four mutations in a gene represents a significant deviation from the inferred Poisson distribution? If so, what statistic was used to assess significance?

Response: According to the inferred Poisson distribution, the mean value for the number of genes predicted to accumulate four mutations is 1.73 (95% CI, 1.28-2.07), while we observed 10 genes each harbored four mutations, a 4.76-fold deviation from the mean. This suggests that most of the genes with four mutations could have been subject to a positive selection. By contrast, we observed 39 genes with three

mutations, which is a 0.78-fold increase from the predicted mean of 21.96 (95% CI, 17.05-24.94) genes with three mutations. This suggests that about half of these 39 genes accumulated three mutations in the absence of positive selection. We therefore chose four mutations as the threshold for identifying genes that had potentially undergone a process of positive selection. The changes in the text that explain this are now found in lines 254-258.

Figure 3: the country annotations are not easily gleaned from the figure. This is particularly challenging when following the presentation on lines 207 and onwards. Perhaps some colored dots on the terminal branches would be better.

Response: We thank the referee for bringing our attention to this problem, but believe that adding colored dots to the terminal branches would be confused with the existing markers indicating non-human isolation sources. Instead, we have altered the figure by adding thicker lines for the terminal branches and hope that this makes it easier to identify the geographic origin of the isolates.

Vegard Eldholm

Reviewer #2 (Remarks to the Author):

In this study, Luo and Xu, et al. sequenced a large collection of *Mycobacterium kansasii* and analyzed the relationship of major clades to previously described subtypes, identified recombination within and between subspecies, examined the relationship between clinical and environmental isolates, performed dating and phylogeographic analyses for a lineage comprising many of the clinical isolates, and identified gene content differences and loci under selection. These analyses substantially contribute to the current literature on *M. kansasii* genomics, and the authors have made the sequencing data publicly available and included detailed

metadata in their supplementary tables.

However, I have two major concerns that should be addressed before publication.

1. How does the sampling effect the conclusions of the phylogeography analysis?

Specifically, I am concerned that a more diverse sampling effort in Europe is leading to the conclusion of a European ancestor.

Response: We agree that differences in sampling sizes and geographic diversity in the different regions could affect the phylogeographic analysis. As shown in Figure 1A, the sampling from Europe and East Asia is relatively large and geographically diverse while samplings from other regions, especially those from South America and Africa are very limited. This limitation is discussed in the revised manuscript in lines 351-353 and the corresponding descriptions have been deleted from the abstract. However, we would like to point out that the geographic origin of the MKMC is not one of the principal findings of this work, which instead emphasizes the robust results concerning the recent origin and global dissemination of the MKMC.

2. How does the culturing methodology affect the conclusions about positive selection?

The authors primarily examine mutations occurring at terminal nodes or at intermediate frequencies in sequencing data for individual samples. I am not currently convinced that these results are not resulting from in vitro adaptation rather than host adaptation based on the details given in the text.

Response: The clinical samples were cultured on Lowenstein Jensen (L-J) slants and multiple colonies that grew on these slants were scraped for DNA extraction. The environmental samples were primarily plated on 7H10 plates from which a single colony of *M. kansasii* complex bacilli were picked and subcultured on L-J slants. Multiple colonies that grew on these L-J slants were scraped for DNA extraction. The above information has been added in the main text in lines 435-439.

We believe that the mutations occurred within the patients rather than during *in vitro* culturing based on the following evidence:

A. None of the single colony subcultures of the water isolates from Australia contained the putative adaptive mutations.

B. In the clinical isolates, taking mutations in *zur* as an example, we detected several isolates that harbored fixed mutations with a frequency of 100% for the mutant allele. As described above, the DNA from the clinical isolates was extracted from the several scraped colonies that grew on L-J slants. Therefore, a fixed mutation represents a mutation that was fixed in all of the scraped colonies that grew from the original clinical specimen. This would be unlikely if the mutation only occurred *in vitro* during culturing. In addition, we observed that the *zur* mutations are very diverse, making it even less likely that the same mutation would occur simultaneously in all of the scraped colonies that grew from a clinical specimen. A more reasonable explanation is that all of the bacteria in the original specimen harbored the same mutation, prior to culturing.

C. Several adaptive mutations in the current study overlapped with the host-adaptive mutations recently found in *M. tuberculosis*, i.e., the *IldD1/2* promoter mutations and the mutations that resulted in the absence of LOS (see references #43 and # 55 in the revised manuscript).

Detailed comments:

54: Since the conclusion that DCT is the likely mechanism of recombination is based on only the lengths of detected recombinant portions of the genome rather than demonstrating that DCT occurs via mating experiments, this sentence in the abstract needs to express some uncertainty about this conclusion (perhaps, “that may be mediated by...”).

Response: This sentence was changed accordingly in lines 55-57.

132-136: Positive Tajima’s D values can also indicate a decrease in population size. Also, the positive values reported here are small (close to zero), so these values might

be the result of neutral evolution. The negative value reported could be indicative of population expansion or a selective sweep. Given the amount of recombination reported by the authors, is a selective sweep acting on one locus likely to affect the whole genome?

Response: Considering the small sample size and/or difference in sampling sources, mentioned by reviewer #3 (below), we now believe that the Tajima's analysis for MKC subspecies other than the *M. kansasii* main cluster (MKMC) was less meaningful and the corresponding results were deleted. Only the MKMC had a sufficiently large number of isolates to yield reliable results and therefore only the Tajima's D analysis for the MKMC was retained and described in lines 213-215.

For the last question, we noticed that most of the variants used for Tajima's D calculation were due to recombination. We therefore calculated the Tajima's D for individual genes by keeping or removing the recombinant fragments and obtained median values of -2.45 and -0.93 respectively, which is consistent with a process of population expansion and/or a selective sweep. In our opinion, a selective sweep on one gene, resulting from either a spontaneous mutation or recombination, should only affect the Tajima's D statistics for that particular gene but not for the whole genome.

139-141: The 16S and 23S for phylogenetic network analysis does not add much here since a network based on the whole genome is also reported.

Response: We have deleted the network and modified the 16s-ITS-23s rRNA alignment in the Supplementary Figure 2.

162-167: It would be helpful for readers to know that Gubbins was used to detect recombination in this analysis since the authors used FastGEAR in the previous paragraph, and Gubbins and FastGEAR detect different recombination events (FastGEAR only identifies recombination between clades/clusters).

Response: The revised manuscript acknowledges the use of Gubbins in line 158.

213: Have the authors considered the effects of sample size on the phylogeographic analysis? Isolates from North America, South America, Africa, Asia (outside of China), and Australia appear to come from 1-2 locations within the continent. Isolates from Europe may be the most diverse because it seems that a geographically diverse sample was obtained based on Figure 1A.

Response: The possibility of sample selection bias was addressed above in our response to your comment #1, above. In lines 351-353 we acknowledge that our findings need to be confirmed by sequencing additional isolates, especially from the areas underrepresented in our sample.

257-258: For mutations that were found to independently evolve, did the authors check to see if these mutations were present in recombinant regions?

Response: To address this concern, we looked at the strains with the mutations that we identified. For potential loss-of-function mutations in *zur* and *tetR1/2*, the large sequence divergence between the recombinant fragments and the reference genome of *M. kansasii* only permitted an examination of nonsense and frameshift mutations. Among the 4 isolates with recombinant fragments in *zur*, we identified no nonsense/frameshift mutations. Among the 70 isolates with recombinant fragments in *tetR1/2*, we identified 14 nonsense/frameshift mutations. We also looked at the convergent mutations in *lldD1/2* promoters and *mce1D*. We found that none of the 8 isolates with recombinant fragments in *lldD1/2* promoters harbored the convergent mutations. Among the 12 isolates with recombinant fragments in *mce1D*, we identified 5 isolates that harbored the *mce1D* 970 G>C mutation. This information has been added in lines 275-276 and lines 307-308.

264-267: Can the authors provide more detail on the culturing and sequencing protocol for these isolates? Are the cultures likely representative of mutations present in patients or are significant bottlenecks expected? Was culturing done the same way for clinical and environmental isolates? Is it possible that mutations found on the terminal branches or at intermediate frequencies are adaptations to in vitro culturing

rather than host adaptation or mapping/sequencing errors?

Response: Please see our response to the Reviewer's major comments #2, above.

310: homologous instead of homogenous?

Response: This has been corrected in line 315.

455: hierBAPS is misspelled

Response: Thank you for catching this. It has been corrected in line 476.

Figure 1A.

It would be helpful to know where the underlying data supporting the prevalence of *M. kansasii* depicted in the map come from (sample collection method, year, etc.).

Response: We have added Supplementary Table 1, which contains the information requested.

Does the size of the symbol representing geographic origin of the isolates have any meaning?

Response: We have modified the size of the symbols in Figure 1A to reflect the number of samples from each geographic region.

Figure 1/2/3.

Could you use a different color scheme for the lineages compared to the isolations source in Figure 1?

Response: We agree that the colors can be confusing, and we struggled with the choice of colors to represent both different regions and bacterial lineages. We tried to make it clearer using different colors schemes, but it is difficult because when there are 13 different colors in the same figure, it will inevitably be hard to discriminate what each color represents. Therefore, we chose to keep the existing color scheme and added an explanation of the colors in the legend.

Figure 3A.

Interestingly, there seems to be some completely clonal clades within lineage 2 in this tree. Is there any evidence that these strains may have lost their ability to participate in DCT (potentially, mutations in the ESX loci)? Or for some reason do not come into contact with other *M. kansasii* strains (potentially human to human transmission)? Or these strains have recently emerged and there hasn't been time for a recombination event to occur (what is the TMRCA for these strains)?

Response: We looked carefully at the ESX loci in these clades but didn't find any loss-of-function (i.e., nonsense or frameshift) mutations in either ESX-1 or ESX-4. And, as presented in the results, we did not find any evidence of human-to-human transmission, not even in the Portland, Australia outbreak. The TMRCA for these isolates is estimated to be around 35-54 years ago, about the same as for the clades that containing frequent recombinations. One possible explanation is that these clades harbored loss-of-function mutations in unknown loci associated with DCT.

Table S3. Potential is misspelled.

Response: Thank you for catching this. It has been corrected.

Reviewer #3 (Remarks to the Author):

I read with interest the manuscript by Luo et al on the evolutionary history and global diversity of *M. kansasii*. This is a very well written manuscript incorporating a very relevant and globally diverse sample. The methods are detailed and sound. In this work the authors confirm the different phylogenetic lineages formed by different subtypes along with the global evolutionary history of the *M. kansasii* main complex and for which the authors can identify different lineages with distinct dispersion patterns. The latter finding is novel and important to provide an adequate background

to the study of the evolving pathogenic ability by this species. The genes proposed to contribute to the virulence/pathophysiology of this species will require further confirmation by in vitro and in vivo studies. Also, the study should have included more environmental isolates as some settings may lack the adequate environmental context.

Response: We agree that further study is needed to confirm the genetic loci that may contribute the success of *M. kansasii*, as mentioned in line 338-339. We also agree that analyzing additional environmental isolates from more diverse areas is essential for accurately tracing the global dispersion of the *M. kansasii* main cluster and for formulating strategies to interrupt its continued transmission and dissemination.

Some comments:

The authors state that at least six *M. kansasii* subtypes exist while the existence of seven subtypes is widely recognized based on PCR-RFLP. Please clarify and acknowledge that no strain from subtype VII was included in the study. The authors did gather a large *M. kansasii* genomic dataset but the absence of subtype VII leaves a gap in the study.

Response: To our knowledge, subtype VII has only been reported one study that classified subtypes based on RFLPs in the *hsp60* gene (Taillard, C. et al. *J Clin Microbiol* **41**, 1240-4, **2003**). With *in-silico* RFLP analysis, we found *hsp60* variants within some of the other subtypes, suggesting that the proposed subtype VII may represent variants in *hsp60* rather than a novel subsp. However, in the strains we analyzed there were recent recombinant fragments with less than 95% identity to *M. kansasii* species I - VI, but even lower percentages of identity to other species of *Mycobacteria*, suggesting that these recombinant fragments could have originated from novel *M. kansasii* subspecies that have not yet been identified, as mentioned in lines 148-152.

The authors start by extrapolating the global diversity of *M. kansasii*, confirm the monophyletic nature of the different subtypes and propose that these should be considered as subspecies rather than subtypes based on genome-wide ANI. The results obtained in this regard are not novel, similar findings have been described by Tagini et al (2019) which in fact propose that the different subtypes of *M. kansasii* should comprise novel species. The authors are not using the currently standing nomenclature, instead propose a new one which in my opinion creates further confusion. The standing nomenclature should be incorporated throughout the manuscript or, alternatively, the authors can explain the reason of not adopting the standing nomenclature based on the evidence emerging from these results. The novel nomenclature proposed by Tagini et al should, nevertheless, be mentioned in the introduction.

Response: We cite the work of Tajini *et al* in the Introduction section and agree that it is preferable to use the standing nomenclature rather than to propose a new nomenclature. Accordingly, we have modified the main text, the figures and tables with the standing nomenclature. We also acknowledge that the classification of *M. kansasii* subtypes into subspecies/species is not novel and have therefore deleted this assertion from the abstract. Nevertheless, the identification of extensive recombination between the species documented in current study adds additional evidence to support the classification of these closely-related species as belonging to an *M. kansasii* complex (lines 314-317).

A low Tajima's D was observed for ssp. I. This was calculated based on sliding windows throughout the genome, although the authors do not provide an explanation on why not using overlapping windows. I believe it would be interesting to evaluate this metric on an individual gene basis. My main concern here is if this diverging value may be driven by the large sample size for ssp. I when compared with the other subtypes due to sampling bias. The SNV distribution is clearly different in ssp. I.

Response: We agree that the diverging values observed between ssp. I with other major subspecies may due to the sampling bias, not only regarding sample size, but

also the sample sources. The isolates of ssp. I and VI were mostly from human hosts, while isolates of ssp. II and III were mostly from water sources. Because our overall sample only contained an adequate number belonging to the MKMC, we believe that a Tajima's D analysis of the other subspecies would not be meaningful. We have therefore deleted the comparisons between subspecies, and only kept the results for the MKMC. The Tajima's D for the MKMC was also recalculated based on individual genes in subspecies I, as suggested, including either all SNVs or only non-recombinant SNVs (lines 213-215, Supplementary Figure 6).

Lines 176-186 – This part is not easily perceived from the figures that are provided or, the annotation of these is insufficient. The fourteen complexes are not clearly identified.

Response: We have changed Figure 2D and added brackets to indicate the 14 complexes.

Regarding the dating analysis, Fig.S7 does not convey the robust temporal signal that is mentioned in Line 211. Please explain. Also, the MRCA date is placed around 1917, the mutation rate is very low and the perspective offered in the discussion is somewhat speculative. What sequences were used as input for BEAST? The core genome or the SNVs obtained by mapping?

Response: An explanation has been added to the legend for Supplementary Fig. 7. The mutation rate is generally very low in slow-growing mycobacteria. For example, in *M. tuberculosis* the rate is estimated to be 1×10^{-8} to 5×10^{-7} nucleotide changes per-site-per year (PLoS Pathog 15, e1008067, 2019), similar to the confidence intervals for the estimates of our observed data, shown in the Supplementary Figure 7B. The full genomic alignment rather than just SNVs was used for BEAST evolutionary analysis, as described in lines 540-543.

In the Methods section, please elaborate further on the collection strategy used for the isolates included in the study along with date intervals for each subset.

Response: The corresponding information has been added in lines 424-435.

Fig.S10 – Please check the spelling of *M. kansasii*.

Response: Thank you for catching this. The spelling has been corrected.

REVIEWERS' COMMENTS

Reviewer #1 (Remarks to the Author):

The authors have handled all my concerns raised.

Reviewer #2 (Remarks to the Author):

In my previous review, I was concerned that the sampling differences between geographic regions were influencing the results of the phylogeography analyses and that the description of the culturing methods was not sufficient. The authors have addressed both of these concerns in the revised version of the manuscript by including additional details on culturing and specifically stating the phylogeography limitations in the discussion section.

I did not catch this the first time I read through the manuscript, but all software used for these analyses are cited with a link instead of an actual citation. I realize that Nature Communications has a suggested reference limit, but it is a disservice to the authors of the software to not properly cite their manuscripts.

I have a few other minor comments from the revised version of the manuscript.

Line 130-132: From the boxplots in Supplementary Figure 1, it doesn't appear that any of the species have a median pairwise SNV difference of 129. More clarification about what is being measured here would be helpful. Perhaps, a nucleotide diversity measure like π would be more interpretable (and comparable to other species/samples).

Line 190: Does ML stand for maximum likelihood? I don't think it has been defined before this use.

Line 216: remove "of"

Line 218: add "s" to strain

Line 282: space missing before 16

Line 332: Does the word special here mean unique to *M. kansasii*?

Line 422: Capitalize S in 16S

Reviewer #3 (Remarks to the Author):

None, I consider that the authors have adequately addressed all points.

REVIEWERS' COMMENTS

Reviewer #1 (Remarks to the Author):

The authors have handled all my concerns raised.

Reviewer #2 (Remarks to the Author):

In my previous review, I was concerned that the sampling differences between geographic regions were influencing the results of the phylogeography analyses and that the description of the culturing methods was not sufficient. The authors have addressed both of these concerns in the revised version of the manuscript by including additional details on culturing and specifically stating the phylogeography limitations in the discussion section.

I did not catch this the first time I read through the manuscript, but all software used for these analyses are cited with a link instead of an actual citation. I realize that Nature Communications has a suggested reference limit, but it is a disservice to the authors of the software to not properly cite their manuscripts.

Re: The links were deleted in the main text and the software used in current study were listed and cited in the Supplementary table 3.

I have a few other minor comments from the revised version of the manuscript.

Line 130-132: From the boxplots in Supplementary Figure 1, it doesn't appear that any of the species have a median pairwise SNV difference of 129. More clarification about what is being measured here would be helpful. Perhaps, a nucleotide diversity measure like π would be more interpretable (and comparable to other species/samples).

Re: Thank you for catching this. The median difference is 1,888 to 3,717 SNVs before removing recombinations (corrected in line 129). Considering the nucleotide diversity/identity within and between MKC species has already been addressed by pairwise genome-wide nucleotide identity, we choose to use SNVs difference here in order to demonstrate that numerous SNVs are introduced by recombinations and the removal of these SNVs significantly decrease the genetic distance between strains for all major species (line 150-153). As you suggest, we added more information in the figure legend to clarify that the comparisons were done between core genomes, which are 2.12Mbp in length.

Line 190: Does ML stand for maximum likelihood? I don't think it has been defined before this use.

Re: Yes, corresponding change has been made in 188.

Line 216: remove "of"

Re: Corrected.

Line 218: add "s" to strain

Re: Corrected.

Line 282: space missing before 16

Re: Corrected.

Line 332: Does the word special here mean unique to *M. kansasii*?

Re: Yes, the word "special" has been changed to "unique" in line 330.

Line 422: Capitalize S in 16S

Re: Corrected.

Reviewer #3 (Remarks to the Author):

None, I consider that the authors have adequately addressed all points.